# Biomechanical and Sensory Feedback Regularize the Behavior of Different Locomotor Central Pattern Generators

**DOI:** 10.3390/biomimetics7040226

**Published:** 2022-12-04

**Authors:** Kaiyu Deng, Alexander J. Hunt, Nicholas S. Szczecinski, Matthew C. Tresch, Hillel J. Chiel, C. J. Heckman, Roger D. Quinn

**Affiliations:** 1Department of Mechanical and Aerospace Engineering, Case Western Reserve University, Cleveland, OH 44106, USA; 2Department of Mechanical and Materials Engineering, Portland State University, Portland, OR 97207, USA; 3Department of Mechanical and Aerospace Engineering, West Virginia University, Morgantown, WV 26506, USA; 4Departments of Biomedical Engineering, Physical Medicine and Rehabilitation, Neuroscience, Shirley Ryan AbilityLab and Northwestern University, Chicago, IL 60611, USA; 5Department of Biology, Case Western Reserve University, Cleveland, OH 44106, USA; 6Department of Neurosciences, Case Western Reserve University, Cleveland, OH 44106, USA; 7Department of Biomedical Engineering, Case Western Reserve University, Cleveland, OH 44106, USA; 8Departments of Neuroscience, Physical Medicine and Rehabilitation, Physical Therapy and Human Movement Sciences, Shirley Ryan AbilityLab and Northwestern University, Chicago, IL 60611, USA

**Keywords:** synthetic nervous system, rat, rhythm generator, pattern formation, perturbation analysis

## Abstract

This work presents an in-depth numerical investigation into a hypothesized two-layer central pattern generator (CPG) that controls mammalian walking and how different parameter choices might affect the stepping of a simulated neuromechanical model. Particular attention is paid to the functional role of features that have not received a great deal of attention in previous work: the weak cross-excitatory connectivity within the rhythm generator and the synapse strength between the two layers. Sensitivity evaluations of deafferented CPG models and the combined neuromechanical model are performed. Locomotion frequency is increased in two different ways for both models to investigate whether the model’s stability can be predicted by trends in the CPG’s phase response curves (PRCs). Our results show that the weak cross-excitatory connection can make the CPG more sensitive to perturbations and that increasing the synaptic strength between the two layers results in a trade-off between forced phase locking and the amount of phase delay that can exist between the two layers. Additionally, although the models exhibit these differences in behavior when disconnected from the biomechanical model, these differences seem to disappear with the full neuromechanical model and result in similar behavior despite a variety of parameter combinations. This indicates that the neural variables do not have to be fixed precisely for stable walking; the biomechanical entrainment and sensory feedback may cancel out the strengths of excitatory connectivity in the neural circuit and play a critical role in shaping locomotor behavior. Our results support the importance of including biomechanical models in the development of computational neuroscience models that control mammalian locomotion.

## 1. Introduction

Mammals can swiftly react to terrain changes during locomotion by dynamically adapting their limb movements. Although the strategies and neural systems that legged animals use to produce stepping motions are still not fully understood, many experiments suggest that the spinal cord and associated peripheral nervous system contribute to the generation and adaptation of gaits [1,2,3]. It is widely believed that neural circuits located in the spinal cord called central pattern generators (CPGs) are important for repetitive motions in animal locomotion, such as those seen in swimming or walking [4,5,6]. These neural circuits can produce patterned output either with or without input from a descending system. However, how the CPG is organized and how descending commands influence its behavior is still not well understood.

Many earlier models of rhythm-generating spinal circuitry assumed a half-center organization for the network, in which two neuronal groups mutually inhibit each other [7,8,9]. The dynamics of the half-center structure have been well studied [10,11,12,13], and some models were successfully applied as robot controllers [14,15,16,17]. However, it is difficult for the half-center model to independently modify both the timing and magnitude of motoneuron activity [18]. Additionally, the half-center model cannot replicate the non-resetting deletions that have been observed in spinal animal preparations [19] in which motoneuron bursts were absent and then subsequent motoneuron bursts reappear at the time they would be expected, had the deletion not occurred.

To enable separate control of CPG timing and MN patterning, Rybak and colleagues introduced a two-layer CPG model [20]. It is composed of a half-center rhythm generator that regulates rhythmic timing and a pattern formation network that distributes and shapes the timing signals to the motoneuron pools. This model provides a degree of separation between the behavior of the frequency control network and the motor patterning network, providing a means for more flexibility in motor activations while maintaining coordinated locomotion. We demonstrated in previous work that when the CPG frequency is increased through a descending drive, the activity level of the pattern formation layer is less affected than that of the rhythm generator layer [21]. Additionally, the two-layer CPG is more adaptive to perturbations, allowing for small timing adaptations at the pattern formation layer that are not experienced at the rhythm generator layer.

The two-layer CPG model has continued to evolve and has been further refined to capture a range of experimental observations through the inclusion of additional structural elements and modeling parameters. For example, recent modeling work has included weak mutual excitation between oscillators in the rhythm generator level of the classic half-center CPG model [21,22]. This connectivity is hypothesized to match observations from several experimental studies [23]. For example, when the inhibition of v1 and v2b interneurons is absent, the flexor and extensor motoneurons can still be synchronously and rhythmically excited [24,25,26]. Although the introduction of this connection increases the explanatory power of this model, how the strength of this connection impacts overall network performance is not clear. Additionally, the sensitivity of the network performance to variations in the strength of the connection between the rhythm-generating and pattern-formation layers, which is fundamental to the performance of this two-layer CPG, has not been examined systematically.

In the present study, we investigate how variations in these connection strengths affect the overall CPG network function. We use a perturbation analysis, evaluating how the phase changes in response to perturbations under different modeling conditions applied at different times in the gait cycle. We employ a similar approach to the phase response analyses used previously to characterize oscillators’ performance in experimental [27,28] and modeling studies [29,30]. Our analysis begins with the examination of the phase response curves of the deafferented CPG, then examines the consequences of coupling the CPG with a simulated biomechanical model that incorporates sensory feedback. The addition of limb mechanics can have important consequences for model performance, either amplifying or suppressing the response of the system to perturbations [31].

We examine several aspects of the performance of this two-layer CPG. We first examine the influence of the hypothesized cross-excitatory connection on the rhythm-generating behavior. Specifically, we analyze how changes in this connection affect the locomotor frequency and phase adjustments in response to perturbations. After that, we vary the synapse strength between the two layers and evaluate the rhythm generator’s ability to phase lock with the pattern formation network at different intrinsic frequencies. The two-layer CPG is then connected to motoneuron pools and used to actuate a simulated biomechanical rat hindlimb, which provides the CPG with sensory feedback. The movement speed is increased through two methods: (1) adding an external drive to stimulate the rhythm generator, as from a direct connection between the midbrain locomotor region (MLR) and the rhythm generator; and (2) increasing the conductance of the mutually excitatory connections between the rhythm generator oscillators, as if neuromodulation were used to increase the synaptic strength. Our results show that, despite trends in differences between the two models during deafferented and air walking, no discernible difference in locomotor behavior is found during ground walking experiments.

## 2. Methods

### 2.1. Neuromechanical Model

The development of the rat hindlimb model was completed in Animatlab [32], a simulation platform for designing neural models and actuating biomechanical bodies with proprioceptive feedback. The configuration and parameter values of the biomechanical model (Figure 1A) are the same as reported in our previous model [21] and the model’s motion is still constrained in the sagittal plane with hinge joints.

To inspect how the simulated models’ reactions to perturbation are influenced by different two-layer CPG designs, the hip joint control network from the previous synthetic nervous system (SNS) was isolated, as shown in Figure 1B. In this network, extensor and flexor Ia feedback, which signals the rate of a stretch of the muscle, synapse onto the Ia interneurons. The flexor and extensor Ib feedback, which signals the tension developed in the muscle, synapses onto the corresponding motoneurons. To eliminate other influences and simplify the problem for this study, only type II feedback was applied directly to the two-layer CPG to coordinate the extensor-flexor timing. This is similar to how increased stretching of hip flexors encourages the initiation of swing in animals, and increased stretching of extensors encourages the initiation of stance [33,34,35].

### 2.2. Neural Model

All neurons in the SNS are non-spiking neurons, where each neuron represents the mean activity of a population of spiking neurons [36,37]. Each node in the network functions as a leaky integrator [29]. Neglecting action potentials increases computational efficiency and greatly reduces runtime [37]. Our model explores how signals propagate through the network and how groups of neurons activate, deactivate, and contribute to network behavior. The dynamics of the neurons are governed by the following differential equations:$$C_{m}\frac{dV}{dt}=G_{m}\cdot \left(E_{r}-V\right)+I_{app}+\sum G_{syn,i}\cdot \left(E_{s,i}-V\right)+G_{Na}\cdot \left(E_{Na}-V\right)\cdot m_{\infty }\cdot h$$
$$G_{syn,i}=g_{syn,i}\cdot min\left(max\left(\frac{V_{pre}-E_{lo}}{E_{hi}-E_{lo}},0\right),1\right)$$
$$z_{\infty }=\frac{1}{1+A_{z}\cdot \exp\left(-S_{z}\;\left(V-E_{z}\right)\right)}$$
$$\frac{dh}{dt}=\frac{h_{\infty }\left(V\right)-h}{\tau_{h}\left(V\right)},\;\tau_{h}\left(V\right)=\tau_{h}\cdot h_{\infty }\left(V\right)\cdot \sqrt{A_{h}\cdot \exp\left(-S_{h}\;\left(V-E_{h}\right)\right)}$$
where *V* is the membrane potential, *C_m_* is the membrane capacitance, *t* is the time variable, *E_r_* is the resting potential of the neuron, and *G_m_* is the leak conductance. *I_app_* is an externally applied current. $\sum G_{syn,i}\cdot \left(E_{s,i}-V\right)$ are currents from synaptic connections, where *E_s_* is the reversal potential of the *i^th^* synapse, $g_{syn}$ is the maximum conductance of the synapse, and *E_lo_* and *E_hi_* are the lower and the upper thresholds of the synapse, respectively. *V_pre_* is the presynaptic neuron voltage. $G_{Na}\cdot \left(E_{Na}-V\right)$ is the persistent sodium current in CPG neurons, which are gated by activation and inactivation of voltage-dependent channels m and h, respectively. The variable *z* represents either m or h, as their equations have the same form, except that m increases monotonically with V but h decreases monotonically with V. Details of all neuronal and synaptic parameters can be found in Appendix A (Table A1 and Table A2).

### 2.3. Bifurcation Parameter

Szczecinski et al. used a bifurcation parameter, δ, to analyze the behavior of half-center CPGs [29]. The bifurcation parameter is defined as:$$\delta =V_{\infty ,inh}-E_{lo}$$
where $V_{\infty ,inh}$ is the equilibrium potential of the inhibited half of the CPG and *E_lo_* is the synaptic threshold it must cross to begin inhibiting the other half. When δ<0, the inhibited half will remain below the resting threshold for all time, establishing a stable equilibrium. When δ>0, the inhibited half can escape from its lower equilibrium and the CPG oscillates. As δ grows larger, the CPG oscillates faster. This previous work showed that the strength of mutual inhibition was related to the value of δ, suggesting a biophysical meaning for this parameter. Thus, Szczecinski’s study [29] explained why δ is effective at predicting the frequency and the phase response to stimuli for half-center oscillators.

The present study expands the analysis from Szczecinski’s work with two additional variables as shown in Figure 2A: (1) the weak excitations conductance $G_{w}$ between the two halves of the rhythm generator; and (2) the descending drive D from the MLR.

The equilibrium potential $V_{\infty }$ occurs when there is no change to the membrane voltage. Thus, when dV/dt=0, one can solve the neuron dynamics equation for $V_{\infty ,inh}$:$$V_{\infty ,inh}=\frac{E_{r}+D+G_{w}\cdot E_{w}+G_{hyp}\cdot E_{hyp}+G_{Na}\cdot E_{Na}\cdot m_{\infty }\cdot h}{1+G_{w}+G_{hyp}+G_{Na}\cdot m_{\infty }\cdot h}$$
where $m_{\infty }$ and h are both dependent on the value of $V_{\infty ,inh}$, as shown above in Section 2.2.

When modeling the CPG circuit, usually the synaptic threshold *E_lo_* is set to equal the resting potential of the neuron, *E_r_*. Hence, the bifurcation parameter δ in this study is described by the following equation, in which each of the potentials has *E_r_* subtracted from it:$$\delta =\frac{D+G_{w}\cdot \left(E_{w}-E_{r}\right)+G_{hyp}\cdot (E_{hyp}-E_{r})+G_{Na}\cdot \left(E_{Na}-E_{r}\right)\cdot m_{\infty }\cdot h}{1+G_{w}+G_{hyp}+G_{Na}\cdot m_{\infty }\cdot h}$$

### 2.4. Overview of Analysis

All analysis of the two-layer CPG in this work is performed through MATLAB (MATLAB 2021a, The MathWorks, Inc., Natick, MA, USA).

First, the experiment in Szczecinski et al.’s study was replicated, with δ, the bifurcation parameter, being modified by varying $G_{w}$, the conductance of the cross-excitatory synapse between the oscillators in the rhythm generator, and D, the descending drive. After that, the effect of the connection strength $G_{c}$ between the two layers was investigated. The two-layer CPG was deafferented with no external inputs and the strength of mutual excitation was adjusted until the rhythm generator oscillated at 1 *Hz*. The 1 *Hz* frequency is close to a slow stride frequency of a rat, making comparisons to in vivo data easier. The intrinsic frequency of the pattern formation layer was varied between 0.5 *Hz* and 2.3 *Hz* by adjusting the mutual inhibition strength. Different connection strengths were then tested for phase locking and phase shift between the two layers.

Phase response analysis was performed on the deafferented two-layer CPG neural model and used to predict the behavior of the mechanical model when integrated, i.e., the neural model generates muscle contractions and receives sensory feedback from the resulting motion. In the deafferented neural model, the rhythm generator’s response to changes in the external drive and the synaptic conductance between the rhythm generators was studied. Then, phase response curves (PRCs) of two rhythm generators, one driven entirely by the external drive and one driven entirely by mutual excitation, were numerically calculated by applying a square pulse stimulus to the RG flexor interneuron. The advancement or delay of the cycle was measured and normalized to the entire cycle. The perturbation current was either −0.5 *nA* or 0.5 *nA* with a duration of 5 percent of the oscillation period, which is consistent with the stimulus in Szczecinski et al.’s study; keeping the same duration here allowed us to compare our results to those described in that previous study. After that, for a two-layer CPG, external stimuli were applied to different layers of the CPG, and the corresponding PRCs were analyzed and compared.

Frequencies of the two-layer CPGs were then increased to 2 *Hz* through (1) adding an external drive to stimulate the rhythm generator; and (2) increasing the conductance of the mutual excitatory connections between the rhythm generator oscillators. PRCs and the phase shift between the two layers were calculated and compared. The faster frequency (2 *Hz*) was chosen to compare how a baseline CPG responds to two different types of stimulation, as though the animal is modulating its speed. In addition, 2 *Hz* is closer to the locomotion frequency of rats observed in previous X-ray treadmill experiments performed by our collaborators [21]. A similar frequency has also been reported in other studies [38].

Finally, each configuration of the network model was incorporated with the biomechanical model. The response of each model to perturbations was tested, and the implications with respect to future and past modeling work are discussed.

## 3. Results

### 3.1. The Effect of Variations in the Strength of Reciprocal Excitatory Connections and Descending Drive on the Performance of the Rhythm-Generating Layer

In Szczecinski’s study [29], the half-center CPG’s oscillation frequency was found to depend on the strength of mutual inhibition. However, this was the case when all other parameters were held constant. The oscillation frequency is actually dependent on everything that sets each neuron’s equilibrium voltage, which includes persistent sodium current parameters, rest voltage, membrane conductance, and other parameters (Section 2.2).

In the present study, we expanded Szczecinski’s results by exploring two additional variables: (1) the weak excitatory conductance $G_{w}$ between the rhythm generator half-centers and (2) the descending drive D from the MLR. We investigated how these parameters influence the behavior of the rhythm generator. For instance, the frequency of 1 *Hz* can be obtained by a rhythm generator with no mutual excitation ($G_{w}$ equal to zero) and external drive equal to 5.86 *nA*, or with a mutual excitation conductance equal to 0.57 *uS* (reversal potential *E_s_* = −40 *mV*) and no external drive. However, the shape of each neural burst underlying these oscillations with similar frequencies is substantially different. As seen in Figure 3, the membrane voltages underlying these two oscillations are different between the two networks, i.e., the rhythm generator that oscillates because of an external drive (Figure 3A) has a larger voltage range and sharper peaks than the rhythm generator that oscillates because of a weak mutual excitation (Figure 3B).

The sensitivity of the two networks to external inputs is also different. Figure 3C,D present the phase response of rhythm generators oscillating at 1 *Hz* (with the same parameters as in Figure 3A,B) when the external current *I_s_* was applied at different phases of the oscillation cycle. As shown in Figure 3C,D, the magnitude and timing of the phase shift differ between these two rhythm generators. To quantify the differences in phase response, the PRCs were numerically calculated by applying a square pulse perturbation current to the RG flexor interneuron and measuring the advancement or delay of the subsequent cycle, normalized to the duration of one unperturbed cycle. The perturbation current *I_s_* was either −0.5 *nA* (Figure 3E) or 0.5 *nA* (Figure 3F) with a duration of 5 percent of the oscillation period. The rhythm generator which was driven by direct stimulus D = 5.86 displays a smaller phase advancement and, thus, is less sensitive to external inputs compared to the other model.

There are further differences in circuit function on the bifurcation parameter δ between changing the strength of the descending drive or the weak excitatory connections. For the rhythm generator driven by constant drive D, δ equals 0.07, whereas, for the network oscillating by mutual excitation $G_{w}$, the δ value is 0.03. As δ represents the sensitivity to phase advancement/delay, it tells us how close the system’s equilibrium state is to the threshold needed to disrupt the equilibrium (and thus start another cycle). For a small δ, the CPG responds immediately to the inputs, and the phase advancement is directly related to when the stimulus current is applied. However, for larger values of δ, the rhythm generator’s sensitivity to inputs is reduced [29].

Figure 4A shows the oscillation frequency of the rhythm generator can be controlled by both reciprocal connections and descending drive, showing that the two parameters ($G_{w}$ and D) have similar effects. For a given descending drive D, an increase in reciprocal connection strength $G_{w}$ will increase the frequency until the neurons’ only equilibrium state is for both neurons to have equal membrane voltage (and thus, they do not oscillate; blue regions in Figure 4A). When D and $G_{w}$ are too large, the system does not oscillate (blue regions in Figure 4A). Figure 4B shows that although different combinations of parameter values might result in the same δ values, the frequencies of the resulting circuits are not equal. Figure 4C shows PRCs for three different parameter combinations for which δ = 0.1. The curves exhibit the same general shape, but their details certainly differ at some locations, especially at the phase delay portion of the curves. In the system driven by descending command D, the phase delay timing is postponed compared to the model driven by reciprocal excitation (i.e., less sensitive).

In summary, a specific oscillation frequency can be obtained for the rhythm generator by different combinations of parameter values but the resulting networks will possess different bifurcation δ values. Since the sensitivity of the network to incoming perturbations is determined by δ, which represents how close the system’s equilibrium state is to the threshold that must be crossed for the CPG to switch states, the networks will have different responses to perturbations.

### 3.2. The Effect of Variations in the Connection Strength between Rhythm Generating and Pattern Formation Layers

In the previous section, it was shown that the synaptic conductances (*G_w_* and *G_hyp_* in Figure 2A) and external drive D influence the rhythm generator’s frequency and phase response to stimuli. Different combinations of parameter values and stimuli lead to PRCs with different shapes and sizes. In the following section, to evaluate the performance of the system, unless otherwise noted, the pattern formation layers are tuned (by varying *G_hyp_*) to the same intrinsic frequencies as the rhythm generators.

#### 3.2.1. Effect of Signal Transmission Strength between Two Layers

When an excitatory synapse (*G_c_* in Figure 1 and Figure 5) connects the rhythm generator (RG) to the pattern formation (PF) layer, the PF layer can become phase locked to the RG layer even when the intrinsic frequency of the PF layer is different than that of the RG layer (Figure 5A). For example, when *G_c_* = 0.05 *uS,* PF layers with intrinsic frequencies ranging from 0.73 to 1.1 *Hz* can be phase locked to an RG that has an intrinsic frequency of 1 *Hz*. When the connection strength *G_c_* is increased, PF layers with a wider range of intrinsic frequencies can become entrained and phase locked to the RG’s frequency. Additionally, different intrinsic frequencies of the PF network result in different phase shifts between the two layers, even when there are no perturbations to the system (Figure 5B). For example, when *G_c_* = 0.1 *uS*, the phase shift between the two layers ranges from 0.1% to 4%. In contrast to the trends seen in the frequency range, the range of phase difference between the two layers is reduced when the *G_c_* connection strength increases. However, when the connection strength is too weak (*G_c_* = 0.05), the system is less likely to phase lock, and therefore, the range of phase differences is also reduced.

Additionally, the coupling strength between the RG and the PF layers affects the response of the network to perturbations. Strong coupling causes the PF layer to strictly follow the signal pattern from the RG, even when the system is perturbed (Figure 6, brown lines). When the connection between the layers is strong (*G_c_* = 0.1 *uS*) and an inhibitory perturbation current *I_s_* is applied at the RG or the PF layer, the phase shift difference between the layers is effectively zero (Figure 6C,D). The PF layer is less affected when the connection strength is reduced (Figure 6, green dashed lines), especially for those stimuli applied at the PF layer. This leads to phase shifts between the RG and the PF layer as shown in Figure 6C,D. These phase shifts reflect how much flexibility the neural system has during perturbed motion. There are no significant differences in the maximum phase shifts when comparing RG perturbations vs. PF perturbations, indicating that the maximum phase shift is primarily related to the connection strength.

While stronger connections can explain the smaller phase shift difference between the two layers, the ratio between the perturbation current and the direct connection conductance (*I_s_*/*G_c_*) also becomes smaller as the connections strengthen. One could surmise that increasing the magnitude of the perturbation in relation to *G_c_* could produce similar phase responses as reducing the coupling strength *G_c_*. To examine this, the PRCs of the same ratio *I_s_*/*G_c_* were also calculated for different *G_c_* (Figure 6E). The results show that the phase response difference for the same *I_s_*/*G_c_* ratio is different, and the reduction in the range of phase difference is largely due to the increase in *G_c_*.

#### 3.2.2. Speed Modulation

Descending commands from the high-level midbrain locomotor region (MLR) have been reported to be responsible for changing gait and posture while an animal is in motion [39]. Our analysis of the rhythm generator examines two mechanisms that could be used to increase the CPG frequency: direct stimulus D of the RG neurons, or neuromodulation of the weak excitatory synapse conductivity $G_{w}$. Further analysis of these two mechanisms on the two-layer CPG indicates that increasing CPG frequency through either mechanism makes the system less sensitive to perturbations (Figure 7A). The maximum range of the phase shift between the two layers for a CPG operating at 2 *Hz* driven by the direct stimulus D is around 32.4% smaller than the one driven through neuromodulation (Figure 7B). This demonstrates that the two-layer CPG driven by the direct stimulus D is less sensitive to perturbation than the one driven by neuromodulation, which is consistent with what we found for the deafferented rhythm generator above, as shown in Figure 3.

### 3.3. Performance of Two-Layer CPG with Simulated Mechanical Model

In previous sections, the experiments were performed by simulating a deafferented two-layer CPG without a simulated mechanical model or sensory feedback. However, feedback plays an important role throughout locomotion, and it is unclear if the inclusion of feedback would give different results in response to perturbations. Therefore, a single joint (hip) model actuated by pairs of antagonistic muscles (Figure 8A) was created to investigate the CPG’s performance under different combinations of parameters during air walking. Type II afferent feedback from the muscles is connected to the RG and PF CPG interneurons through inhibitory connections (Figure 8B). This feedback reduces the cross-inhibition within the CPG, enabling the suppressed half to escape sooner. The leg model is suspended in the air, and parameters are set to produce ‘air-walking’ movements of the hip joint. Synapse properties were hand-tuned to produce similar maximum joint angles as seen in the walking rat (approximately 20 degrees in the flexion direction and 40 degrees in the extension direction).

We then recorded the response of the neuromechanical system to external torque perturbations applied to the joint at different points of the locomotor cycle. Figure 9 presents the joint angle vs. time for the single joint model with a frequency of 2 *Hz*. The torque perturbation is applied at 0.5 s and ends at 1 s. When the external torque is small (Figure 9A), both models recover after the perturbation ends. However, the model driven by direct stimulus recovers slightly sooner and returns more closely to its unperturbed step timing. When a large perturbation torque is applied, one step cycle is deleted (Figure 9B), and the simulated hip joint receiving the external drive recovers joint motion faster (blue line, Figure 9B) than the network with weak excitatory connections (red line, Figure 9B). The simulated hip joints in Figure 9 do not fully recover to their original step timing due to limited feedback and alteration of the rhythm generator timing.

In the next experiment, we investigated if the simulated air-walking results above would translate to simulated locomotion with ground contact. Therefore, we used the same neural parameters in the walking simulation model with two hindlimbs described above (Figure 1A). We then placed the simulated hindlimb model on the ground and applied Ib feedback from the ankle extensor to the RG and PF CPG extensor interneurons. Figure 10 shows the perturbed motions for the hip joint in our ground walking model walking at 2 *Hz*. For both models, D excitation or $G_{w}$ neuromodulation, the hip joint quickly recovers to its original step timing after both small and large perturbations. Neural models using either mechanism exhibit powerful stability against the disturbance, and it is difficult to distinguish the difference in kinematic offset for these two mechanisms (Figure 10D).

## 4. Discussion

This work investigates some of the details of the hypothesized two-layer CPG model thought to underlie mammalian-legged locomotion [18,20,22] and how the two-layer CPG and its associated biomechanical model react to perturbations when the locomotion frequency is changed. This study builds upon our previous work [21] and investigates how neural parameters in the CPG cooperatively generate patterned signals needed for locomotion. These analyses provide guidelines for future computational neuroscience designs for mammalian locomotion modeling studies.

The sensitivity analysis of the rhythm generator reveals that both the external drive (e.g., from the MLR) and the weak excitatory strength between oscillators influence the oscillation frequency. Increasing the excitatory strength proportionally alters the bifurcation parameter, δ, similar to previous work that demonstrated increases in δ due to increases in direct stimulus [29]. However, the neural activity for these two mechanisms is different even when the oscillation frequency is the same. Half-centers in the model driven by the direct drive have a stronger onset of activity and a flatter plateau when compared to the model driven by mutual excitation. The phase response curves of the rhythm generators in response to perturbations are also different; the model driven by the direct stimulus is less sensitive to external perturbations as it has a higher value of δ given the same intrinsic frequency.

The connection strength between the two layers determines whether the two layers will lock phases and if so, the phase delay between the layers. A high connection strength forces the pattern formation network to replicate all phase shifts and phase resetting that occur in the rhythm generator. This replicates the type of connection that has been observed at the hip joint, where the joint position is closely related to the swing-stance transition [33,34,35]. However, this behavior does not apply to all the joints as it is not very flexible when it is necessary to adapt patterned signals in response to perturbations (e.g., sensory feedback altered by uneven terrain) and can lead to stability problems during locomotion. For example, the ankle joint is known to have a more flexible pattern as foot placement relies significantly on feedback from muscle spindles during walking [35]. The results here show that a weaker connection strength can provide this additional phase shift flexibility between the two layers.

Increasing the frequency of the two-layer CPG generally decreases the system’s sensitivity to external inputs. This holds true for both the deafferented neural circuit, as well as for the neuromechanical model performing air walking. When connecting the CPG with a physical body and type II feedback, trends in the PRCs analyzed in the deafferented model are directly reflected in the behavior of the neuromechanical model. The simulated hip joint driven by the direct stimulus is less disturbed during a small torque perturbation and recovers faster when a step cycle is totally deleted during air walking.

In contrast, when either model is implemented in a neuromechanical model of two hindlimbs walking on the ground, most of these differences disappear. Both neural models exhibit powerful stability against the disturbance and there is no functional difference in step timing for these two mechanisms. These results suggest that the precise behavior of the deafferented neural circuit may be of little importance to how the whole system performs in the real world. This finding is consistent with observations based on sea slug feeding behaviors [40], where the isolated ganglia motor patterns are very individualized and sensory feedback reduces individuality. Indeed, for any controller, neural or artificial, to be robust and capable, it must not be highly sensitive to internal parameters, and must properly incorporate sensory feedback to shape and coordinate overall behavior.

Moreover, biomechanics plays a significant role in stabilization. A study from Matrangola et al. [41] suggests that musculoskeletal properties might be important in perturbation corrections: increases in body mass and inertia are beneficial to balance recovery, as it increases the resistance to a perturbation. Additionally, mechanical interactions, such as the stretching of elastic tissue, start instantaneously after a perturbation. These so-called ‘preflexes’ [42] occur much faster than sensory information can be processed and usually serve as the first line of defense against perturbations [43]. These previous studies clearly demonstrate that peripheral biomechanics can significantly improve stability, and our work provides further support for this important observation. Finally, our results also show that sensory feedback pathways within the spinal cord can provide further stabilization of locomotor gaits, demonstrating that locomotor stability is not strictly a feature of central neural circuits but likely results from the combined contributions from several elements in the peripheral motor system [44].

What causes the air-walking and ground-walking models to behave differently? There are two changes made from air walking to ground walking: the ground contact and Ib feedback from the ankle extensor, which is the load sensory feedback. We tried to build a ground walking model without Ib afferent. However, as the inclusion of Ib feedback makes ground walking more stable, we still do not know the answer. More attention should be paid to investigating the influence of load feedback in the future. Would the system become more sensitive to perturbations if the feedback strengths were reduced? Or is it the shape of the load feedback or the ground contact timing that stabilizes the phase transition from swing to stance? Further experiments exploring a range of perturbations to investigate time-torque relationships and the strength of sensory feedback might help resolve some of these issues.

There is another limitation in this study that may be important. We only implemented II and Ib feedback into the CPG. As joint motion profiles are used to tune the neural parameters, presumably, the strong velocity sensitivity afferent would be similar for air-walking and ground-walking models. However, the Ia feedback might also strongly affect CPG phase relations. Hence, the air-walking and ground-walking models might be expected to perform differently. Therefore, further inspection of how Ia feedback influences CPG phase relations is also necessary for future studies.

The results of this study suggest that the strengths of excitatory connectivity do not have to be fixed precisely for stable walking. This suggests focusing on more detailed biomechanical models to investigate whether additional muscles or muscle groups provide benefits to the stability of the model as suggested by some previous work with biarticular muscles [45]. Biarticular models require a more complex neural system than monoarticular models. This present study suggests how parameters might be set in the CPG of a biarticular muscle model. This work also provides guidelines for future mammalian locomotor modeling by constraining parameter value ranges and reducing computational costs during parameter searches.

## Figures and Tables

**Figure 1 biomimetics-07-00226-f001:**
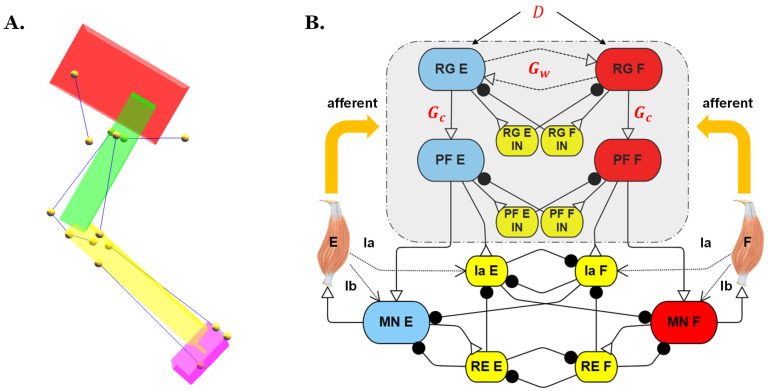
(**A**) Biomechanical model of a rat hind limb from our previous work [21]. Each leg segment is modeled as a rigid body (pelvis in red; femur in green; tibia in yellow; and tarsus in purple). Muscles are indicated by blue lines, and yellow dots are the origin, insertion, and intermediate points. (**B**) Joint control network; parameters labeled in red font were those investigated in the present study. D: Supra-spinal drive; E: Extensor; F: Flexor; IN: Interneuron; RG: Rhythm generator; PF: Pattern formation; RE: Renshaw cell; and MN: Motoneuron.

**Figure 2 biomimetics-07-00226-f002:**
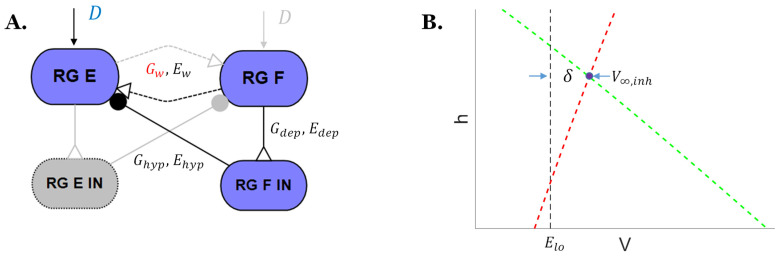
(**A**) Schematic of the detailed neural connections for the inhibited rhythm generator extensor neuron in Figure 1b. (**B**) δ in the phase space of the rhythm generator. The red dashed line is the *V*-nullcline and the green dashed line is the *h*-nullcline.

**Figure 3 biomimetics-07-00226-f003:**
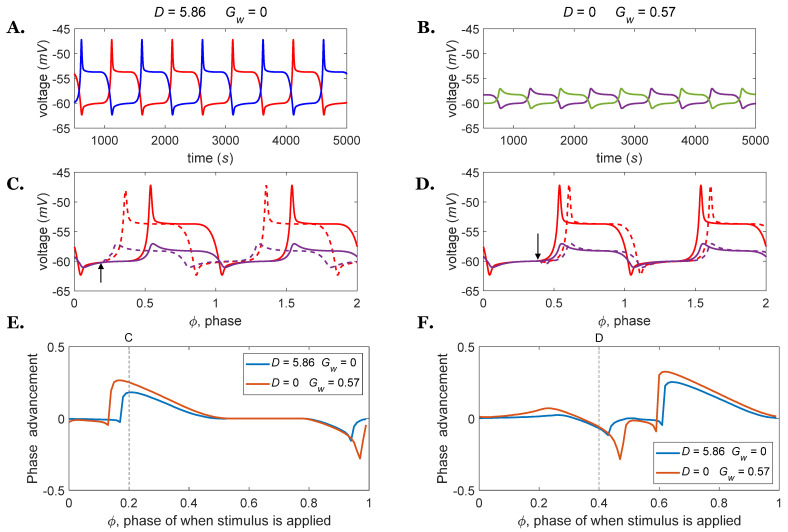
(**A**) Rhythm generators oscillating at 1 *Hz* driven by a descending drive, 𝐷 = 5.86. (**B**) Rhythm generators oscillating at 1 *Hz* driven by weak excitatory connections, $G_{w}$ = 0.57. (**C**,**D**) Comparisons between perturbed (dashed lines) and nominal (solid lines) membrane voltages for rhythm generator extensors are shown in A (red lines) and B (purple lines); (**C**) the *inhibitory* perturbation current *I_s_* = 0.5 *nA* was applied to the RG flexor interneuron at φ = 0.2 with a duration of 5 percent of the oscillation period (arrow); (**D**) the *excitatory* perturbation current *I_s_* = 0.5 *nA* was applied to the RG flexor interneuron at φ = 0.4 with a duration of 5 percent of the oscillation period (arrow). (**E**) PRCs for the rhythm generator driven by D = 5.86 or $G_{w}$ = 0.57 at 1 *Hz* when perturbed by inhibitory stimuli of the same magnitude and duration as shown in part (**C**). (**F**) PRCs for the rhythm generator driven by D = 5.86 or $G_{w}$ = 0.57 at 1 *Hz* when perturbed by excitatory stimuli of the same magnitude and duration as shown in part (**D**).

**Figure 4 biomimetics-07-00226-f004:**
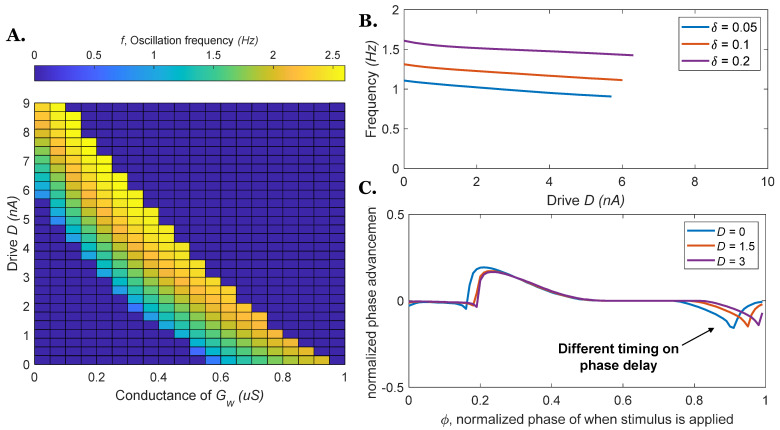
The frequency of the rhythm generator and phase response curves vary as a function of neural parameters (descending drive D or coupling conductance $G_{w}$). (**A**) Various combinations of D and $G_{w}$ can control the oscillation frequency (presented as a color map) of the rhythm generator. (**B**) Different combinations of parameters might result in the same δ values, but the frequencies will be different. (**C**) The phase response curves for δ = 0.1 with different parameter combinations. The phase response curves are relatively similar, however, differences clearly exist.

**Figure 5 biomimetics-07-00226-f005:**
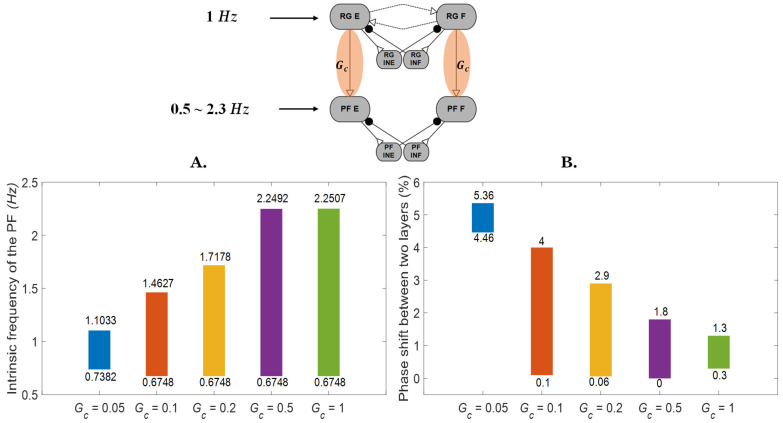
The rhythm generator phase locks with the pattern formation layer. (**A**) The range of pattern formation layer frequencies that phase locked to the rhythm generator is related to the connection strength of *G_c_*. (The top number reflects the maximum frequency that could phase lock, whereas the bottom number reflects the minimum frequency that could phase lock.) Outside of the ranges indicated by the bars, the two layers could not phase lock. (**B**) As connection strength increases, the range of phase differences between the pattern formation layer and rhythm generator is reduced. (The top number represents the maximum phase shift as a percentage of the normalized cycle, whereas the bottom number represents the minimum phase shift as a percentage of the normalized cycle.) However, when the connection strength is too weak (*G_c_* = 0.05), the system is less likely to phase lock, and therefore, the range of phase differences is also reduced.

**Figure 6 biomimetics-07-00226-f006:**
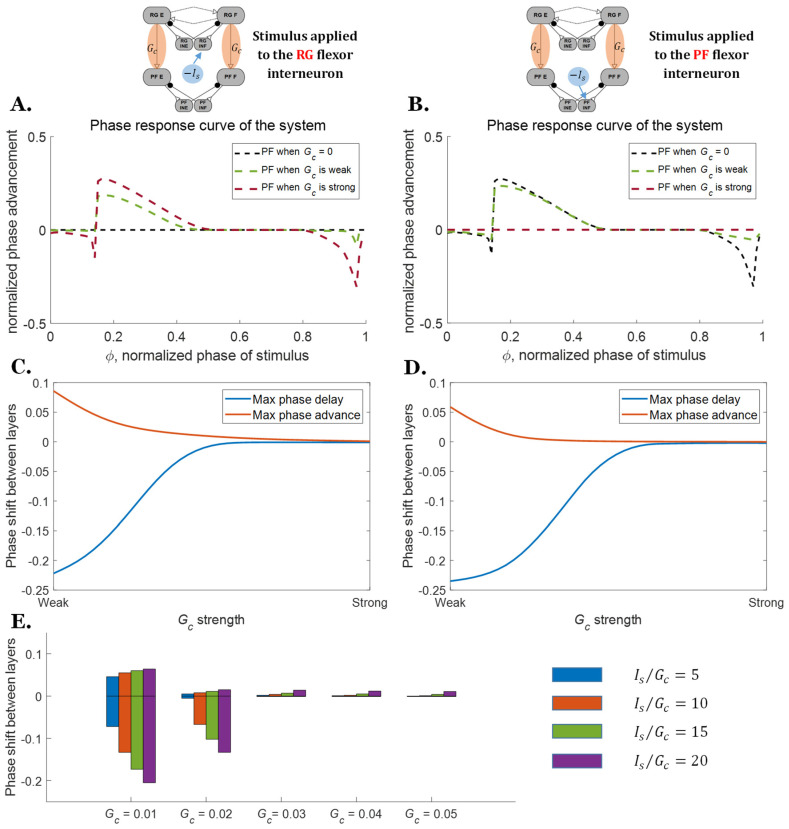
PRCs of the two-layer CPGs when perturbed with the stimulus of different magnitudes or at different phases. (**A**) PRCs of the two-layer CPG when the inhibitory stimulus is applied to the RG flexor interneuron. (**B**) PRCs of the two-layer CPG when the inhibitory stimulus is applied to the PF flexor interneuron. (**C**) Maximum possible phase shifts between the two layers when the stimulus is applied to the RG layer. (**D**) Maximum possible phase shifts between the two layers when the stimulus is applied to the PF layer. (**E**) Range of possible phase shifts between the two layers for different $I_{s}/G_{c}$ ratios when the stimulus is applied to the RG layer.

**Figure 7 biomimetics-07-00226-f007:**
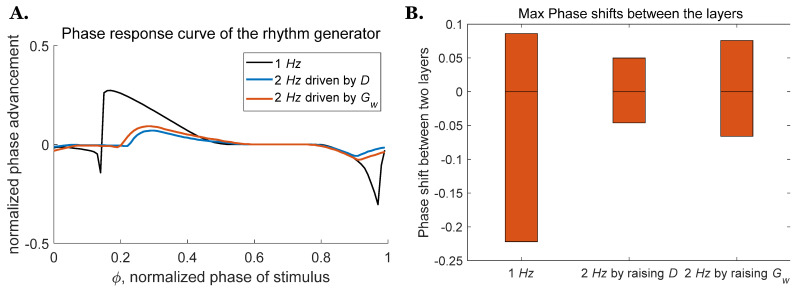
PRCs and maximum phase shift for the rhythm generator as frequency increases. (**A**) PRCs for the rhythm generator at 2 *Hz* driven by D or neuromodulation of $G_{w}$. (**B**) Range of phase shift between the two layers when oscillation frequency is increased.

**Figure 8 biomimetics-07-00226-f008:**
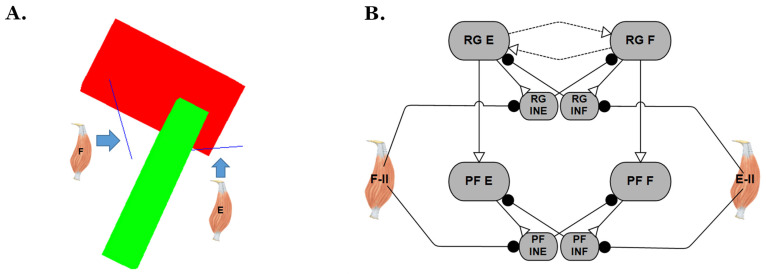
Schematics for the single joint biomechanical model. (**A**) Single joint (hip) physics model shown in the neutral position (θ = 0). (**B**) Type II afferent feedback of the muscles was connected to CPG interneurons (CPG part of Figure 1B) on the opposite side through inhibitory connections.

**Figure 9 biomimetics-07-00226-f009:**
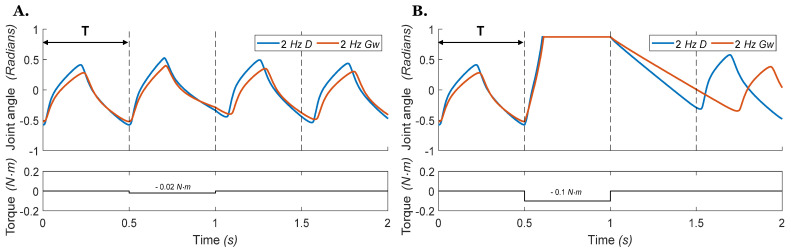
Hip joint angles vs. time during stepping with an external perturbation torque applied to the joint during ‘air-walking’ in which the leg does not touch the ground. The torque is applied on the femur at the start of the second stance phase at 0.5 s and ends at 1 s, with a duration of one step cycle. The blue 2 *Hz* D line indicates that the CPG is operating at 2 *Hz* and driven by the direct stimulus D. The red 2 *Hz*
*G_w_* line indicates that the CPG is operating at 2 *Hz* by neuromodulation of the weak excitatory synapse conductivity $G_{w}$. Dashed lines indicate when the transition from extension to flexion would occur if there was no perturbation. (**A**) Given a small torque perturbation (−0.02 Nm), both models recover after the perturbation ends. (**B**) A large torque perturbation (−0.1 Nm) causes the deletion of an entire step cycle.

**Figure 10 biomimetics-07-00226-f010:**
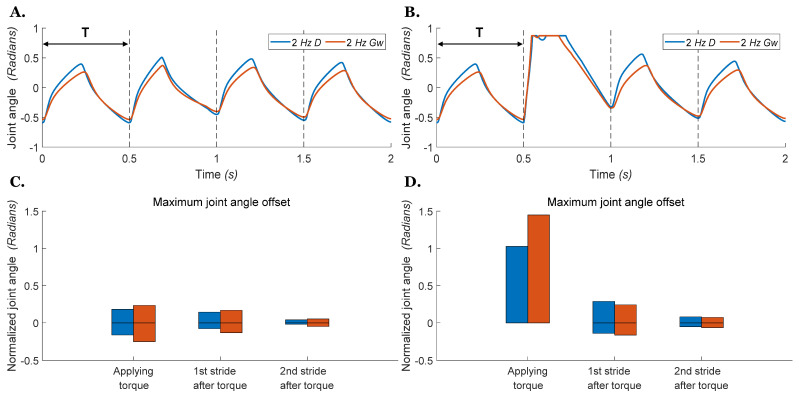
Hip joint angle vs. time during stepping with an external torque applied to the joint during ground walking. The blue 2 *Hz*
*D* line indicates that the CPG is operating at 2 *Hz* and driven by the direct stimulus D. The red 2 *Hz*
*G_w_* line indicates that the CPG is operating at 2 *Hz* by neuromodulation of the weak excitatory synapse conductivity $G_{w}$. The torque is applied on the left femur at the start of the second stance phase with a duration of one step cycle. Dashed lines indicate when the transition from extension to flexion would occur if there was no perturbation. (**A**) Small torque perturbation (−0.02 Nm) as in Figure 9A. (**B**) Large torque perturbation (−0.1 Nm) as in Figure 9B. (**C**) Bars present the range of joint offset from the nominal profile during and after applying a small external torque as in (**A**). (**D**) Bars present the range of joint offset from the nominal profile during and after applying a large external torque as in (**B**).

## Data Availability

Not applicable.

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
