# Peer review of "Biomechanical and Sensory Feedback Regularize the Behavior of Different Locomotor Central Pattern Generators"

_biomimetics, 2022, doi:10.3390/biomimetics7040226_

Round 1
Reviewer 1 Report
Biomimetics review
This manuscript describes a series of in silico experiments designed to explore the role of descending (or direct) input onto, and excitatory interconnections between the RG layers of a two layer CPG model that incorporates perturbations applied to different components of the model and a biomechanical model of the limb. This manuscript is one in a series that is building our understanding of how interconnections between and among the various components of circuitry models influences their behavior and response to external changes. Despite the mathematical complexity, the authors do a good job of explaining the experiment and what it means in a way that non-modelers can appreciate the outcomes/results and their potential significance.
Several important points are made. They show that both direct drive onto the RGs and excitatory coupling between RGs can result in oscillatory activity that changes with strength of input, but they arrive at similar frequencies using different patterns of circuit activity. Next, they demonstrate how the connection strength between the RG and PF layers dictates the entrainment of the PF by the RG. They then show that the stability of the rhythmic activity in the face of a perturbation is much greater for the direct drive model than for the excitatory coupling model. Finally, they demonstrate, by attaching a biomechanical model of a hindlimb to their circuitry model, that perturbations during air stepping have a more dramatic impact on the rhythmic output that perturbations during loaded stepping, and that perturbations have a smaller impact as the frequency increases (due to input drive). Overall, these are interesting observations that should help to push the modeling field forward and help in vivo studies to address more relevant hypotheses.
Major issues:
The statement in the discussion “These results suggest that the precise behavior of the deafferented neural circuit may be of little importance to how the whole system performs in the real world.” is extremely important and I think bears further explanation/exploration. Please dive deeper and, if possible, revisit this concept/issue in the summary paragraph.
Minor issues:
The exploration of a joint torque perturbation on the system is interesting but limiting the perturbation to 0.5 seconds at 2 different torques is rather unsatisfying and leaves the reader wanting more (which is better than wanting less, I suppose). However, it would be a great addition to reveal if there is a time-torque relationship to the sensitivity of the systems.
In the earlier experiments that did not include a biomechanical model you chose 1Hz for the primary CPG frequency, but in the later experiments (biomechanical model) you chose 2Hz without any rationale being given. Please explain.
Figures:
Figure 10; the x-axis labels look like one sentence. Perhaps make them 2 lines.
Reviewer 2 Report
This modeling experiment tested the effects of increased direct excitation (D) to the rhythm-generating layer or RG (modeling descending drive) and changing the weak excitatory synapse connectivity (Gc) between the extensor and the flexor RG half-centers (modeling spares excitatory IN interactions identified between flexor (F) and extensor (E) RG neurons). A previously used (by the same lab) neural model is used to model the two-layered CPG organization networks. The main findings are that interactions of weak excitatory IN connections and direct excitatory drive to the RG layer control oscillatory PF frequency in the range of 0.5 to 2 Hz. The connection strength between the RG F-E centers alters the phase-shift range at which the two layers could be locked- the weaker the Gc, the wider the phase-shift range. Moreover, a weaker Gc enabled a wider range of possible phase-shift between the RG and PF layers when external input–like perturbations to the RG layer was modeled. The phase shifts between layers were also limited by raising D so being superior to the outcomes when keeping Gc low. These results were applied in a biomechanical model (single-joint with F-E muscles) in order to examine responses to perturbations during walking with and without ground contact. The differences between Gc vs. D parameter-related changes in response to perturbations were much less in the biomechanical model. These were interpreted as a possibility for mechanical entrainment and sensory feedback to reduce the impact of differences in “neural parameters” within the CPG.
The term “neural differences” on Line 32 and 497 should be changed to “strengths of excitatory connectivity” as that was the parameter to which the results hold.
Otherwise, the paper is well written, and all figures are appropriately constructed.
Minor issues and suggestions for improvements:
A definition of the abbreviation “PRC” is needed, and in general, a list of abbreviations could be inserted to help the reader.
Line 253: “As shown in the picture” – insert Fig and panel reference for a picture.
Fig. 4, 6, 7 and 8: Titles for the figure are missing from the legends of these figures.
The duration of the perturbations was set to 5% of the period. What was the rationalization of this value based on?
